# Dermatocosmetic Emulsions Based on Resveratrol, Ferulic Acid and Saffron (*Crocus sativus*) Extract to Combat Skin Oxidative Stress-Trigger Factor of Some Potential Malignant Effects: Stability Studies and Rheological Properties

**DOI:** 10.3390/pharmaceutics14112376

**Published:** 2022-11-04

**Authors:** Delia Turcov, Ana Simona Barna, Alexandra Cristina Blaga, Constanta Ibanescu, Maricel Danu, Adriana Trifan, Anca Zbranca, Daniela Suteu

**Affiliations:** 1“Cristofor Simionescu” Faculty of Chemical Engineering an Environmental Protection, “Gheorghe Asachi” Technical University, 73 Prof. Dimitrie Mangeron Street, 700050 Iasi, Romania; 2“Petru Poni” Institute of Macromolecular Chemistry, 41A, Grigore Ghica Voda Alley, 700487 Iasi, Romania; 3Faculty of Pharmacy, “Grigore T. Popa” University of Medicine and Pharmacy, Universitatii Street no. 16, 700115 Iasi, Romania; 4Faculty of Medical Bioengineering, “Grigore T. Popa” University of Medicine and Pharmacy, Universitatii Street no. 16, 700115 Iași, Romania

**Keywords:** emulsions, ferulic acid, microbiological control, rheological properties, resveratrol, saffron extract, oxidative stress, skin protection

## Abstract

The increasing incidence of skin diseases, against the background of increased pollution, urbanism, poor habits in lifestyle, work, rest, diet and general medication, led to the development of products with a protective effect. These new types of dermatocosmetic preparations ensure maximum benefits with minimal formulation. Antioxidants are, nowadays, ingredients that stand out with a proven role in skin protection from oxidative stress and its effects. Thus, research has shown that light-textured formulas, quickly absorbed into the skin, with optimum hydration and protection against excessive free radicals, uphold the skin integrity and appearance. This article aims to evaluate essential criteria for a newly marketed product: stability, rheological properties and microbiological characteristics of oil-in-water emulsions based on a mixture of 3% resveratrol 0.5% ferulic acid and 1mL alcoholic extract of Saffron. The tests led to the conclusion that O / W dermatocosmetic emulsions, based on 3% resveratrol and 0.5% ferulic acid, or also 1mL alcoholic extract of Saffron, show resistance to microbiological contamination, good rheological properties (viscoelastic behavior, structural stability, acceptable shearing behavior) that reveal satisfactory texture and high physical stability during storage. These results encourage the transition to dermatological testing as the final stage in considering a new commercial product.

## 1. Introduction

Dermatocosmetic care has reached high levels in recent years, due to multiple factors, interrelated to each other. The desire to have healthy and beautiful skin is old and persists unaltered. In addition, modern times have brought to everyone, women and men, young and old, more information, and therefore, better education in this regard. Another category of causes refers to the increase in the incidence of skin diseases, against the background of increased pollution, urbanism with its effects, lifestyle, work, rest, nutrition and general medication habits [1,2].

The skin is exposed to both external and internal factors, and due to genetic predispositions that increase the personal risk of developing skin diseases, or in the case of pre-existing conditions, moderate or severe forms of skin sensitivity, dermatitis, acne, rosacea, premature aging appear [3,4,5,6]. Seasonality also places its mark on the appearance and comfort of the skin, through manifestations triggered, maintained or aggravated by environmental factors specific to each season.

The skincare ritual changes with age, seasonal change, general body condition and evolution of possible skin conditions, all taking into account the needs of the skin and the objective of care or treatment. In choosing personal dermatocosmetic care products we take into account a multitude of criteria designed to lead to the use of the most suitable product for each patient.

A trend in modern dermatocosmetics is to offer minimalist products, with ingredients selected according to clear performance criteria, and targeting skin problems with maximum efficiency [7,8]. Obtaining consistent, real, multiple benefits through minimal loading of the skincare formula is considered a success, so the approach of the formulation is an essential step in the birth of a dermatocosmetic product [9].

Among the common therapeutic objectives in skin care is protection from harmful factors, and in the patient’s compliance aspect is the optimal sensoriality when using the product.

In terms of the selection of active ingredients, antioxidants are becoming more and more important in skin care formulas, due to the constant need to protect the skin from oxidative stress [2,5]. A multitude of factors in the current environment, together with modern lifestyle, with the omnipresence of medication, processed foods, smoking, etc., lead to an excess of free radicals in the body’s skin, against which natural regulating mechanisms are overcome. The damage to the skin is due to the deterioration of the membrane lipids and proteins and the cellular genetic material [1,4].

Important factors, such as the large surface of the cutaneous organ, massive direct exposure to UV radiation and pollution and the toxicity of some pollutants from food, air and medication that manifest in the skin, make this organ the main target of some species of free radicals that can increase pathological risks, of which the malignant one is a powerful danger.

Thus, formulas with light textures, which are quickly absorbed into the skin and provide the necessary hydration and protection from excess free radicals, ensure the integrity of the skin and its tissues, as well as the even, fresh and luminous appearance of the skin.

Oil-in-water type emulsions are suitable for many types of skin [10] and are more stable. Generally, emulsions are very attractive presentation forms for patients, due to their pleasant and comfortable sensation after skin contact, easy application and quick absorption in the skin. Other significant advantages of emulsions are the possibility of offering active ingredients in high concentrations, the option of layering skincare or make-up products and are very suitable for sensitive skin or diverse dermatological conditions.

Resveratrol and ferulic acid are among the most studied and efficient antioxidant ingredients, widely used in topic cosmetic products. Their biological effects are multiple but the free-radical scavenging activity stands out [11,12,13]. The plant extract of Saffron (*Crocus sativus*) is a preparation rich in flavonoids and polyphenols, whose strong antioxidant capacity is combined with that of resveratrol and ferulic acid, increasing the performance of the final skin protection emulsion.

The introduction of a new dermatocosmetic preparation requires a series of analyses regarding its quality and safety profile. Among the necessary quality characteristics are physical–chemical stability (phase separation, coalescence, precipitation, color changes, rheological behavior, etc.), lack of microbiological contamination and tolerability by human tissues.

In this context, the purpose of the article is to evaluate relevant parameters in terms of stability, antioxidant activity and microbiological contamination of the obtained emulsions that contain as active components resveratrol (RV) (3%), ferulic acid (FA) (0.5%) and vegetal extract of Saffron.

## 2. Materials and Methods

### 2.1. Materials

#### 2.1.1. Active Compounds

RV (>90% purity) was purchased as an extract from Aroma-Zone (France). It was obtained from Japanese knotweed, *Polygonum Cuspidatum* on butylene glycol support. FA (98% purity) was purchased from Ellemental (Romania) and was obtained by extraction from the roots of *Ferula assafoetida* L. The mixture’s, RV (3%) and FA (0.5%), properties were studied in our previous paper [14].

Saffron extract (with 1.659 mg/mL total polyphenols and 7545 μg/mL flavonoids [13]) was obtained by solid–liquid extraction using the hot reflux method under the following conditions: ratio solid/liquid 1:16, 60 min extraction time, and concentration of extraction reagent- EtOH 50% [15].

#### 2.1.2. Obtaining Emulsions with Dermatocosmetic Applications

Six types of commercial O/W were prepared based on the studied mixture RV (3%) and FA (0.5% (*w*/*w*)), and the mixture of RV (3% (*v*/*w*)), FA (0.5%), and vegetal extract of Saffron (6.66% (*v*/*w*)). These emulsions were differentiated by using three bases (A, B, C) with different chemical compositions (emulsifier, co-emulsifiers and stabilizer) (Figure 1). The choice of the main ingredients in the composition of the bases A, B and C was made in order to increase the performance of the product through a high permeability and to ensure a pleasant texture and optimal sensory characteristics for the patient/customer.

The final emulsions (Table 1) were obtained according to a detailed protocol in our previous paper [15]. According to it, after the aqueous and oily phases were heated to 75 °C, they were mixed (oily over aqueous) using a rotor-stator (ESGE Zauberstab M 160 G Gourmet, Switzerland) which operates at 15,000 rpm. After the final mixture was cooled to 40 °C (on an ice bath), the active ingredients (mixture of resveratrol and ferulic acid), the preservative and finally a quantity of gel obtained by dissolving HA oligo and HMW in Hamamelis flower water. For further studies, 15 g samples were weighed and then packed in glass containers of suitable capacity which were kept in optimal conditions.

### 2.2. Methods

#### 2.2.1. Stability Evaluation

Evaluating the quality of a new product also involves a series of analyzes that monitor its stability. In this regard, according to the quality standards, a series of analyzes were carried out for the prepared emulsions in this regard under certain conditions [16,17,18], such as: the organoleptic assessments, the pH determination, the separation of the phases under the action of centrifugal force, the determination of the conductivity, the appreciation of homogeneity and microbiological control. The samples subjected to any analysis were considered at room temperature. Stability determinations (pH, conductivity and centrifugation test) were performed at different times during the preparation phase, depending on the type of analysis, but in general, the following time intervals were used: immediately after preparation, after 7 days and after a month from preparation and storage under normal conditions (temperature of 25 °C, in airless containers made of material that prevents the penetration of light rays).

#### 2.2.2. pH Determination

For the determination of the pH values of dermatocosmetic emulsions, a digital pH Meter (Hanna Instrument, Mauritius) was used, and the protocol presented in our previous work was respected [15].

#### 2.2.3. Phase Separation

For centrifugation, the test was performed using 5 g of the sample which was introduced into a model XC-Spinplus (Shanghai, China) for 30 min at 25 °C and 3000 rpm, the 5 g sample of emulsion and the experimental conditions were: 30 min at 25 °C and 3000 rpm [13].

#### 2.2.4. Microscopic Images

To study the morphology and homogeneity of the samples, we used the analysis at the microscopic level. For this purpose, samples were taken from the respective emulsions after 7 days and, respectively, one month after preparation and storage under normal conditions. A binocular microscope Optika B-159 (OPTIKA S.r.l., Ponteranica (BG), Bergamo, Italy), magnification—1000×, was used.

#### 2.2.5. Conductivity Measurements

The conductivity measurements were performed using a portable Hanna Instruments-type conductometer (Nusfalau, Romania), and emulsion samples were stored at an ambient temperature of 25 °C for 30 days.

#### 2.2.6. Microbiological Control

The microbiological analysis has been realized by following an adapted protocol from ISO 18415:2007 [19], for the total aerobic mesophilic microorganisms (total aerobic microbial count and total yeast and mold count). All the manipulations were realized under a laminar flow hood (Steril Helios MI2754b, Milano, Italy). The samples were analyzed 24/48 h and 3 months after their preparation. The containers and the working surface were disinfected with an aqueous mixture of 70% ethanol (*v*/*v*) and 1% HCl (*v*/*v*). A total of 1 g of each sample was collected aseptically, a sterile screw-cap tube containing 1 mL sterile Tween 80 plus five 5 mm glass beads and mixed thoroughly. The total tube volume was adjusted to 10 mL with sterile (autoclave Biobase BKM-Z18N, Shandong, China) MLB—Modified Letheen Broth (8 mL) for the 10^−1^ dilution. From this, 0.1 mL were inoculated into a 90 mm Petri dish with solidified sterile MLA—Modified Letheen Agar, using the spread plate technique to facilitate the recognition of different colony types. All the samples were analyzed in triplicates simultaneously with a medium sterility check (control blank—without product sample). The plates were incubated for 3 to 5 days, at 30 ± 2 °C, using a Biobase Incubator BOV-V35F (Shandong, China), prior to counting the number of colonies using the colony counter (Scienceware ULB-100, Shandong, China) to estimate the total viable colonies growing on each plate (values used for determining the mean of three plates). The total microbial count is reported as CFU/g, accounting for the dilution factor, and the detection parameter is reported as present or absent/g. Microorganisms isolated from the Petri dish were grown on selective media: Mannitol Salt Agar, MacConkey Agar, Eosin Methylene Blue, Cetrimide Agar and Brilliant Green Phenol Red Agar in order to examine pathogenic strains: *Staphylococcus aureus*, *Escherichia coli*, *Enterobacter aerogenes*, *Pseudomonas aeruginosa* or *Salmonella*.

#### 2.2.7. Antioxidant Activity

The assessment of total phenolic and flavonoid contents in dermatocosmetic emulsions

In order to obtain the functional dermatocosmetic emulsion extracts, 0.5 g of each emulsion was extracted with ethanol (10 mL) using a magnetic stirrer (30 min, at room temperature), followed by filtration (0.22 µm pore diameter) (the methodology was adapted after [20,21]). The total phenolic content (TPC) and total flavonoid content (TFC) were determined following previously described protocols [22,23] and by using a SpectroStar Nano Microplate Reader (BMG Labtech, Ortenberg, Germany).

The TPC was determined using the Folin–Ciocalteu method. Briefly, 50 µL of emulsion extract was mixed with 100 µL Folin–Ciocalteu reagent and vigorously mixed. After 3 min, 75 µL of 1% sodium carbonate solution was added and the mixture was incubated for 2 h at room temperature in the dark. A blank was prepared by adding the sample (50 µL) to the Folin–Ciocalteu reagent (100 µL) without sodium carbonate. The absorbances of the sample and blank were read at 760 nm in a 96-well microplate. The absorbance of the blank was subtracted from that of the sample and the total phenolic content was expressed as milligrams of gallic acid equivalents (mg GAE/g cream).

The TFC was determined using the aluminum chloride method. Briefly, 100 µL of emulsion extract was mixed with 100 µL of 2% aluminum chloride and the mixture was incubated for 10 min at room temperature. A blank was prepared by adding the sample solution (100 µL) to methanol (100 µL) without aluminum chloride. The sample and blank absorbances were read at 415 nm in a 96-well microplate. The absorbance of the blank was subtracted from that of the sample and the TFC was expressed as milligrams of rutin equivalents (mg RE/g cream).

#### 2.2.8. Antioxidant Activity Assessment of Dermatocosmetic Emulsions

##### 2,2-Diphenyl-1-picrylhydrazyl Radical Scavenging Assay

The assay was performed following a method previously described [22], with slight modifications. Thus, 50 µL of emulsion extract was added to 150 µL of 2,2-diphenyl-1-picrylhydrazyl (DPPH) 0.004% methanol solution. After a 30 min incubation at room temperature in the dark, the absorbance was determined at 517 nm. DPPH radical scavenging activity was expressed as milligrams of Trolox equivalents (mg TE/g emulsion).

##### 2,2′-Azino-bis(3-ethylbenzothiazoline) 6-sulfonic Acid Radical Scavenging Assay

The assay was performed following a previously described method [20], with some changes. ABTS•^+^ was generated after mixing 7 mM 2,2′-azino-bis(3-ethylbenzothiazoline) 6-sulfonic acid (ABTS) solution with 2.45 mM potassium persulfate (1:1, *v*/*v*). The mixture was left to stand for 12–16 min at room temperature in the dark. At the start of the determination, the ABTS solution was diluted with methanol to reach an absorbance of 0.700 ± 0.02 at 734 nm. Then, 30 µL of emulsion extract was added to 200 µL ABTS solution and strongly mixed. After a 30 min incubation at room temperature, the absorbance was read at 734 nm. The ABTS radical scavenging activity was expressed as milligrams of Trolox equivalents (mg TE/g emulsion).

#### 2.2.9. Rheological Tests

Rheological tests were performed on a modular Physica MCR 501 rheometer (Anton Paar, Graz, Austria) equipped with a Peltier temperature control system. For all rotational and oscillatory measurements, we used the geometry of parallel plates 25 mm in diameter. All isothermal experiments were performed at constant temperatures (25 °C and 35 °C—skin temperature). Reproducibility was checked for all the rheological tests on three samples from each dermatocosmetic emulsion [24].

#### 2.2.10. Rheological Measurements

Flow curves were recorded in the 0.01 to 100 s^−1^ domain at a constant temperature.

The amplitude sweep is used to determine the limit of the linear viscoelastic (LVE) range. Here, the oscillation frequency is kept constant (ω = 10 rad/s), while the oscillation amplitude (γ) is varied (between 0.01 and 100%).

The frequency sweep tests were performed with constant amplitude (γ = 0.1%, in the linear viscoelastic range) and the oscillation frequency is varied (between 0.1 and 100 rad/s). All measurements were carried out at a constant temperature: 25 °C and 35 °C.

Dynamic temperature sweep tests were induced after equilibration at the initial temperature (10 °C). The samples were heated from 10 °C to 50 °C at a 0.5 °C/min rate at a constant frequency of 1 Hz and a constant strain in the linear viscoelastic region (γ = 0.1%).

Time sweep tests were performed at constant parameters: temperature 25 °C, frequency 1 Hz and amplitude 0.1% (in the linear viscoelastic range).

## 3. Results

The emulsions were prepared using, as active ingredients, a mixture containing 3% RV and 0.5% FA, respectively, 3% RV, 0.5% FA and 1mL Saffron alcoholic extract. The mixture RV-FA and the vegetal extract of Saffron, which aimed to increase the antioxidant capacity of emulsion, were characterized in our previous works [14,15].

The prepared emulsions were characterized in terms of rheological behavior and stability to possible microbiological contamination during storage or the use of dermatocosmetic preparation [16,17,18]. All the experimental determinations that followed the stability of the emulsions were performed in triplicate.

The pH determinations showed values between 4.65–5.005 in the case of emulsions from the A-base series; 4.7–5.001 for B-based emulsions and 4.7–5.004 for C-series emulsions.

Conductometric analysis was used to determine how the active substance (plant extract) that was confined in the primary emulsion’s inner phase behaved. It is well known that the active ingredient is free to travel in the external aqueous phase and that the effects will be less long-lasting the more an electrolyte is released. If the conductivity values rise while the product is being stored, this may be because the active ingredient is diffusing, the internal and aqueous phases are coalescing, or the oil film is being destroyed by osmotic pressure and internal aqueous phase leaking [16,17,18]. The measurements’ findings demonstrate the examined emulsions’ stability throughout time. Conductivity determinations showed values between 0.21 and 0.48 mS.

The results of the analyses regarding the stability of the aqueous and organic phases under the action of some mechanical forces are presented in Table 2.

The microscopic imagines obtained in the case of studied emulsions are shown in Table 3.

### 3.1. Microbiological Control

Dermatocosmetics can be contaminated with different microorganisms (bacteria or fungi) during the production process or after its commercialization (during its use by the consumer), both of which can cause health hazards. The components that are part of the cosmetic product can be consumed by certain microorganisms, favoring their development [25]. In order to obtain an adequate product from a microbiological point of view, the manufacturer must take all measures necessary to ensure that the marketed product complies with the quality standards, fully compliant with health and safety regulations [26]. For the production of a stable cosmetic product, it is necessary that the manufacturer has implemented an integrated quality management system, with reference to the raw material and an appropriate formulation of the product, compliance with hygiene standards in the production facility, checking containers used for packaging and, last but not least, a validated addition of preservatives [27]. After production, the added preservatives and responsible use by the consumer keep the product safe [26].

Dermatocosmetic emulsions contain, in general, favorable nutrients which support microbial growth; therefore, the maintenance of good quality is extremely important: they should be free from pathogen microorganisms and the total aerobic bacterial load should be below limits in order to not to cause any skin infections [28]. Usually, microbial contamination has two different starting points: the production process, including ingredients, tools and bottle filling; and, secondary, during usage by the patient. That is why a strict hygiene protocol and correct preparation of all working components were carried out. Subsequently, the microbiological evaluation of the final product has been completed. Table 4 summarizes the microbiological test results with the number of measured colony-forming units per gram of sample.

### 3.2. Antioxidant Activity

The studies on the antioxidant activity of the prepared and analyzed emulsions aimed, in the first phase, to determine the total amount of compounds with antioxidant action, respectively, polyphenols (TPC) and flavonoids (TFC), after which the determination of the antioxidant activity of the emulsion. The results are suggestively presented in Table 5.

### 3.3. Rheological Tests

#### 3.3.1. Flow Curves

The characteristic pseudoplastic behavior of the sample at both temperatures was observed (Figure 2) [29].

#### 3.3.2. The Amplitude Sweep

The storage modulus (G’) is higher than the loss modulus (G”) indicating a solid-like behaviour (Table 6, Figure 3) [30].

#### 3.3.3. The Frequency Sweep

Another rheological test very important for cosmetic emulsions characterization is the frequency sweep test at room and skin temperature. In the viscoelastic linear range, the storage modulus (G’) is higher than the loss modulus (G”) over the frequency range for all cosmetic emulsions at both temperatures (Figure 4), and the dispersed particles do not form sediment within the cosmetic emulsions [31,32,33].

#### 3.3.4. Dynamic Temperature Sweep Tests

The dynamic moduli of cosmetic emulsions decrease with an increase in temperature (Figure 5).

#### 3.3.5. Time Sweep Tests

Oscillatory time sweep tests were performed to estimate the structural stability of the prepared cosmetic emulsions. These tests (Figure 6) revealed that samples are stable in time.

## 4. Discussions

All the pH values obtained for dermatocosmetic analyses show good compatibility with the pH that can be recorded at the level of the epidermis.

The values obtained for conductivity in the case of the analyzed emulsions demonstrate their stability over time.

The results about the stability of the aqueous and organic phases under the action of some mechanical forces presented in Table 2. underline good compatibility of the great majority of them and a good intact appearance, following the action of these forces on the emulsions, not being realized a separation of them (with the exception of sample B3 where the separation of the liquid at the base of the vial is observed and sample C2 where the formation of thin foam is observed in the upper part of the vial).

The microscopic characterization, with the help of optical microscopy, highlighted the homogeneous structure of the emulsions, with globules of different sizes depending on the base that was the basis of the formulation (Table 3), respectively, 4–28 μm in the case of emulsions from series A and B and 2–8 μm in the case of emulsions from the C series. This fact can be attributed to the different composition of the C base (the presence of two distinct compounds from the “green” category: sucrose stearate and soy lecithin with the role of emulsifier and co-emulsifier) compared to bases A and B.

The organoleptic analysis of the emulsions followed the color, the texture and the smell, and highlighted each time for these parameters: color: bright white, slightly shiny; texture: homogenous, firm, light/non-greasy, with no lumps detected after more than 72 h; and odor: of specific components. In the case of emulsions based on C, a slightly yellowish color is observed (Table 2), distinct from the other six samples, a color that can be attributed to the co-emulsifier “soy lecithin”.

All of these results about the stability of the considered products show that all the dermatocosmetic emulsions tested and the bases that were the basis of their production show a high degree of stability to the action of physical stimuli.

### 4.1. Microbiological Control

The analyzed emulsions exhibited very low bacteria contamination (FDA limit of <10^3^ cfu/g), proving that the manufacturing process guarantees microbiological safety and compliance. The yeast contamination was associated with the samples containing 0.5% FA and 3% RV and 1ml Saffron extract, probably due to low contamination of the raw material, since in other formulations (with Saffron extract and base A and B) no contamination was noted. The bacteria strains obtained on the positive samples were analyzed (grown on selective media) and were not *Staphylococcus aureus, Escherichia coli, Enterobacter aerogenes, Pseudomonas aeruginosa* or *Salmonella*, thus proving that the dermatocosmetic product was suitable for use.

### 4.2. Antioxidant Activity

The results presented in Table 5 clearly show that in the case of emulsions with the bases A and C, the introduction of Saffron extract had the effect of increasing the antioxidant activity. This is observed in the case of both determination methods used. These results are, thus, a valuable source of information that, together with the results of the other determinations, will constitute criteria for selecting the emulsions that will follow the course of the in vitro tests.

### 4.3. Rheological Tests

The rheological tests provide quantitative information on a product’s attributes, by measuring the flow and deformation behavior of a sample [31,34]. The understanding and the control of the rheological behavior are very important for the fabrication, transportation, storage and application of emulsions [35]. Cosmetic emulsions are multicomponent mixtures containing stabilizers, thickening agents, surfactants, co-surfactants and other ingredients [29] with rheological parameters depending on the properties and interaction of the components [35]. The application and acceptance of cosmetic emulsions are dependent on the rheological behavior of the final products [36]. The information about the rheological properties of cosmetic creams is very important to develop consumer-acceptable final products [29,37].

#### 4.3.1. Flow Curves

Yield stress is very important to investigate the value of applied force needed to cause the emulsion to flow [31,38,39]. The yield stress values for cosmetic emulsions (τ_0_ = 0.7–41.32 Pa, Casson method) may be attributed to the existence of emulsion stabilizers and thickening agents in products. A yield-stress parameter is correlated with the sensory feel that consumers experience during usage [40,41]. In general, the yield stress of cosmetic emulsions should have a high value to not flow out of the container due when the recipient is placed in an upside-down position, but should not be too higher to offer significant resistance to flow during application on the human skin [42]. A shear-thinning behavior (Figure 2) was observed in dermatocosmetic emulsions because the changes in microstructure components appear. When increasing the shear rate, the number of entanglements between components is decreased, and the structure becomes deformed and destroyed [43]. The component segments would be aligned in the direction of flow, and the shear viscosity is decreased.

#### 4.3.2. The Amplitude Sweep

Xanthan gum is a polymeric material that exhibits a structured network in solution (‘gel-like’ behaviour). The oscillation measurements are used to obtain information about the viscoelastic behaviour of xanthan gum in aqueous solutions.

The storage modulus (G’) is higher than the loss modulus (G”) indicating a solid-like behaviour (Table 6) [30].

As shown in Figure 3, the linear viscoelastic region (LVR) (γ_LVE_ = 0.1% for both temperature) gives information about the structure of the emulsions, i.e., the magnitude of the LVR indicate the degree structure of the creams [31,44].

#### 4.3.3. The Frequency Sweep

The storage and loss moduli (Figure 4) do not cross at low values of angular frequencies, indicating that emulsions have a solid-like behavior and non-sticky sensation on the skin [43]. The high value of storage modulus is an essential condition for time-dependent structural stability [31,45,46].

The linear viscoelastic properties of cosmetic emulsions may be correlated with the interactions (cross-linking, entanglement and aggregation) of components. The linear viscoelastic behavior could be used to evaluate the storage stability of cosmetic emulsions at rest [47,48,49,50]. Cosmetic emulsions must have a predominant solid-like behavior in order to keep the shape and a stable structure for a long time [29].

#### 4.3.4. Dynamic Temperature Sweep Tests

The storage modulus is always higher than the loss modulus over the entire range of temperatures because the emulsion stabilizers and thickening agents determine the intensification of the interaction of components.

In samples A, A1 and A3 (Figure 5a) the internal structure formed by the xanthan gum, polyglyceryl-6 stearate and polyglyceryl-6 behenate can determine the decrease in molecular mobility and the slight increase in dynamic moduli in the range temperature 25–35 °C [29,51,52].

#### 4.3.5. Time Sweep Tests

These tests (Figure 6) revealed that samples are stable in time. For some cosmetic emulsions, a slight increase in the dynamic moduli in time was observed. This can be explained by the possible evaporation effects leading to sample concentration.

## 5. Conclusions

Three types of dermatocosmetic emulsions were prepared, starting from three different bases with the idea of giving them different consistency and texture, and as active ingredients, a mixture of RV (3%) and FA (0.5%), respectively, a mixture of RV (3%), FA (0.5%) and alcoholic Saffron (*Crocus sativus*) extract.

Following the analyses and tests performed on the emulsions obtained, it was found that they have good stability, compatibility of components over time and resistance to microbiological contamination.

Dermatocosmetic emulsions show varying production stress values, which can be correlated with the skin sensation that consumers experience during use. The samples show shear thinning behavior. This behavior is correlated with the ability to spread when dermatocosmetic emulsions are applied to human skin. Oscillatory testing confirms the viscoelastic behavior of these dermatocosmetic emulsions. The samples have a solid behavior, a property that can explain the self-storage stability of dermatocosmetic emulsions.

Large studies and substantial documented articles show the link between oxidative stress and significant carcinogenic risk [53]. Particularly, in skin cells, oxidative stress is a major contributor to mutagenesis and carcinogenesis [54]. Thus, the role of exogenous antioxidants is a certain, essential one, scientifically proven for over two decades [55]. Based on these premises, the emulsions studied in this work can be considered of a certain prophylactic value for skin with high carcinogenic risk. Furthermore, a specific investigation performed is an essential step in the formulation and marketing of a high-quality product.

The results presented here are important achievements that allow the continuation of the necessary tests to complete these formulations. Further, the next investigations will focus on selected formulas that showed high stability and impeccable behavior under the conditions required by specific protocols.

## Figures and Tables

**Figure 1 pharmaceutics-14-02376-f001:**
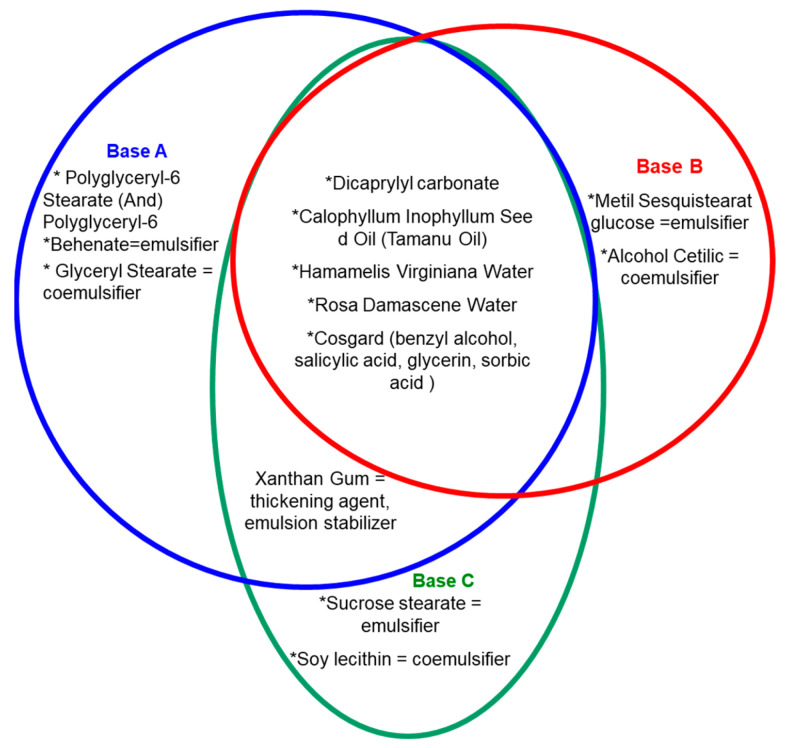
The chemical compounds of the bases involved in the formulation of the three commercial dermatocosmetic emulsions.

**Figure 2 pharmaceutics-14-02376-f002:**
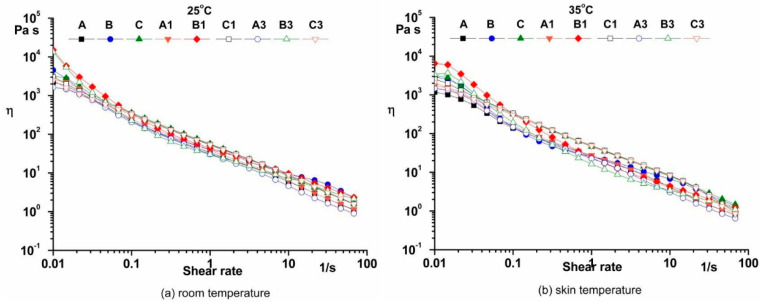
The flow curves at 25 °C (**a**) and 35 °C (**b**).

**Figure 3 pharmaceutics-14-02376-f003:**
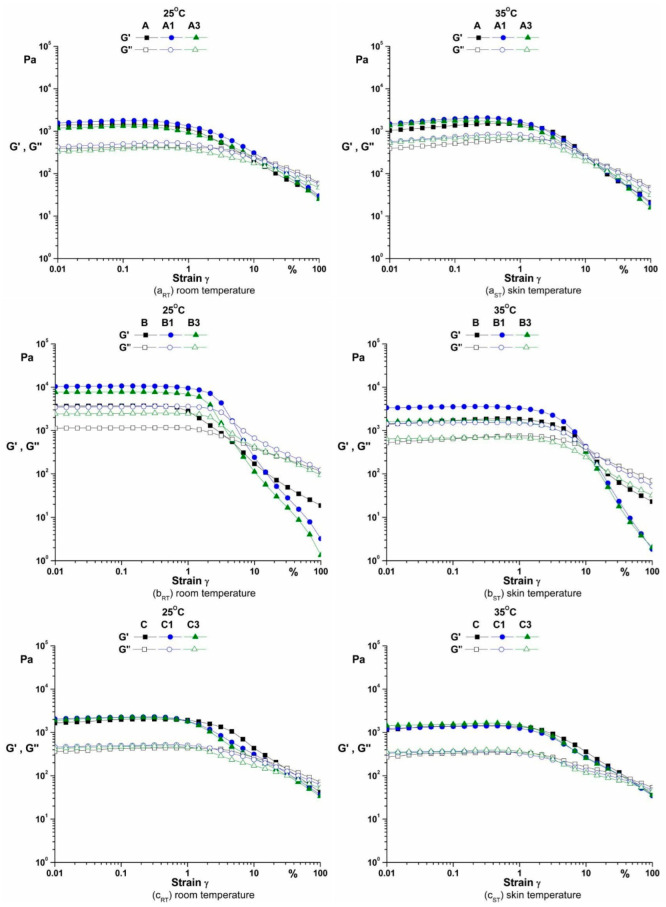
The amplitude sweeps of emulsions.

**Figure 4 pharmaceutics-14-02376-f004:**
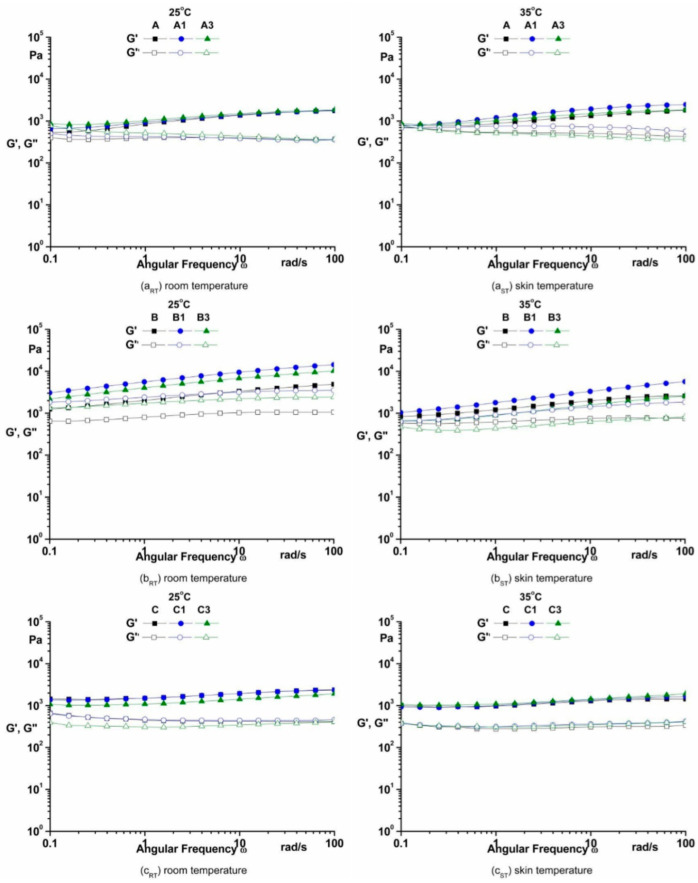
The frequency sweeps of emulsions.

**Figure 5 pharmaceutics-14-02376-f005:**
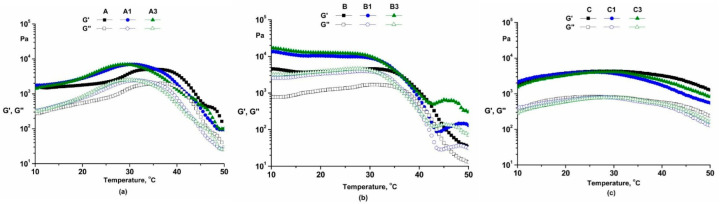
Temperature test in oscillation mode for emulsions A, A1, A3 (**a**), B, B1, B3 (**b**) and C, C1, C3 (**c**).

**Figure 6 pharmaceutics-14-02376-f006:**
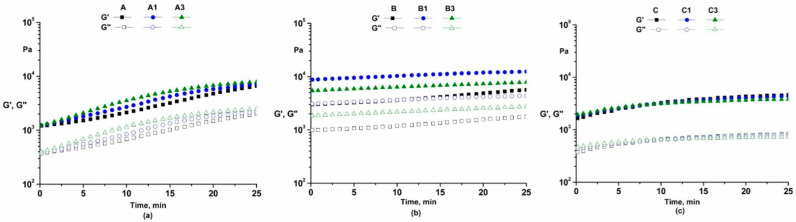
Time test in oscillation mode for emulsions A, A1, A3 (**a**), B, B1, B3 (**b**) and C, C1, C3 (**c**).

**Table 1 pharmaceutics-14-02376-t001:** The type of obtained emulsions.

Base	Phase, % (*w*/*w*)		Active Substances, % (*w*/*w*)
	Dispersed (O)	Continuous (W)	
**A**	28	67.3	4.7
**B**	29	67	4
**C**	37	58.3	4.7

**Notation**: **A1**—Base A + 0.5% FA + 3% RV; **A3**—Base A + 0.5% FA + 3% RV +1 mL Saffron extract; **B1**—Base B + 0.5% FA + 3% RV; **B3**—Base B + 0.5% FA + 3% RV + 1 mL Saffron extract; **C1**—Base C + 0.5% FA + 3% RV; **C3**—Base C + 0.5% FA + 3% RV + 1 mL Saffron extract.

**Table 2 pharmaceutics-14-02376-t002:** The results of the centrifugation stability tests.

Emulsion Sample								
A	A1	A3	B	B1	B3	C	C1	C3
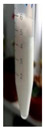	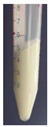	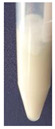	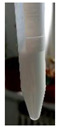	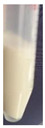	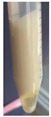	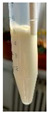	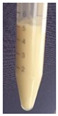	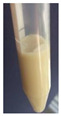

**Table 3 pharmaceutics-14-02376-t003:** Optical microscopy image of studied emulsions.

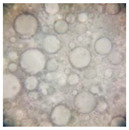	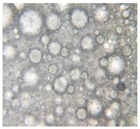	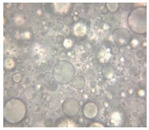
Base A	A1 emulsion	A3 emulsion
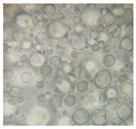	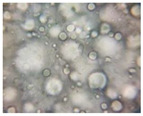	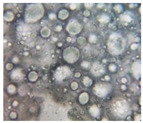
Base B	B1 emulsion	B3 emulsion
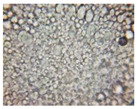	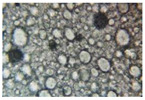	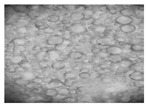
Base C	C1 emulsion	C3 emulsion

**Table 4 pharmaceutics-14-02376-t004:** The result of microbiological tests 3 months (plate reading after 24/48 h) after inoculation for the sample belonging to the forms including the A, B and C basis.

Dermatocosmetic Emulsion	Total Viable Microbial Count, UFC/g	Total Viable Bacterial Count, UFC/g	Total Yeast and Fungi Count, UFC/g
Base A	0	0	0
A1	0	0	0
A3	10	0	**10**
Base B	0	0	0
B1	10	10	0
B3	10	0	**10**
Base C	0	0	0
C1	0	0	0
C3	0	0	0

**Table 5 pharmaceutics-14-02376-t005:** TPC, TFC and antioxidant activity of dermatocosmetic emulsions.

Sample	TPC (mg GAE/g Emulsion)	TFC (mg RE/g Emulsion)	DPPH (mg TE/g Emulsion)	ABTS (mg TE/g Emulsion)
A1	21.32 ± 0.40	0.17 ± 0.01	4.38 ± 0.02	10.85 ± 0.08
A3	6.25 ± 0.38	0.10 ± 0.01	5.63 ± 0.09	12.77 ± 0.26
B1	20.68 ± 1.35	0.29 ± 0.02	9.41 ± 0.05	13.14 ± 0.14
B3	8.24 ± 0.81	0.65 ± 0.05	7.90 ± 0.01	11.65 ± 0.03
C1	23.31 ± 0.95	0.06 ± 0.00	7.74 ± 0.06	12.72 ± 0.22
C3	21.96 ± 0.88	0.27 ± 0.02	8.57 ± 0.11	13.48 ± 0.15

Data are presented as mean ± standard deviation (SD) of three determinations. Abbreviations: ABTS—2,2′-azino-bis(3-ethylbenzothiazoline) 6-sulfonic acid; DPPH—1,1-diphenyl-2-picrylhydrazyl; GAE, gallic acid equivalents; RE—rutin equivalents; TE—trolox equivalents; TFC—total flavonoid content; TPC—total phenolic content.

**Table 6 pharmaceutics-14-02376-t006:** The storage modulus G’ and the loss modulus G” at strain 0.1% for both temperatures 25 °C and 35 °C.

Samples	Strain (ɣ = 0.1%)			
	G’ (Pa)		G” (Pa)	
	T = 25 °C	T = 35 °C	T = 25 °C	T = 35 °C
A	1460	1370	415	512
B	3710	1700	1150	647
C	2000	1400	428	333
A1	1780	1960	500	745
B1	10700	3550	3540	1510
C1	2250	1380	498	362
A3	1320	1760	383	679
B3	7790	1690	2480	676
C3	2210	1540	472	366

## Data Availability

Not applicable.

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
