# Peer review of "Dermatocosmetic Emulsions Based on Resveratrol, Ferulic Acid and Saffron (*Crocus sativus*) Extract to Combat Skin Oxidative Stress-Trigger Factor of Some Potential Malignant Effects: Stability Studies and Rheological Properties"

_pharmaceutics, 2022, doi:10.3390/pharmaceutics14112376_

Round 1

Reviewer 1 Report

Delia Turcov et al. reported a novel Dermato-cosmetic emulsion for oxidative stress applications based on active components resveratrol (RV), ferulic acid (FA), and extract of Saffron. The authors evaluated stability, rheological measurements, and microbiological contaminations.

Overall, the work is novel. However, the authors need to improve the results and discussion section for a better appeal to the readers of Pharmaceutics. The manuscript is suitable for publication after addressing the following concerns.

Comments:

1.     The title is overly ambitious. The authors did not perform any antioxidant studies. I suggest the following title limiting the scope of the manuscript data.

 “Dermatocosmetic Emulsion Formulation of Resveratrol, Ferulic Acid and Saffron (Crocus sativus) Extract : Stability Studies and Rheological Properties.”

2.     “The plant extract of Saffron (Crocus sativus) is a preparation rich in flavonoids and polyphenols, whose strong antioxidant capacity is combined with that of resveratrol and ferulic acid, increasing the performance of the final skin protection emulsion.”

Authors should explain this. Is this hypothesis or a result? Based on what data do you think the performance is increasing?

References required.

3.     Mention the composition of “vegetal extract of Saffron (1mL)” 1mL doesn’t mean much to the reader. Clarify this in the manuscript. What dilution? what composition? How did you analyze it?

4.     Figure 1: redraw the figure. circle borders should not overlap the text. Use arial font 12-14 for better visibility. Read https://pubs.acs.org/doi/10.1021/acs.chemmater.6b00306

5.     Table 1: reader cannot reproduce your emulsion based on this. Mention composition how many grams each for how many mL.

All ingredients of each formulation must be mentioned in the manuscript. Including base compositions.

6.     I was not able to find this reference. Provide DOI in revision.

Turcov, D., Barna, A.S., Apreutesei (Ciuperca), O.T., Puitel, A.C,, Suteu, D., Preliminary processing of floral bio-residues of saffron (Crocus sativus l.) as an innovative resource for development of high added-value cosmetic products. BioResource, 2022, 17(3), 4730-4744.

7.     Figure 3-7 are extremely confusing. Please use A, B,C for each figure and clearly write in the legend what each figure is. The reader should know what exactly he/she is looking at. Modify the discussion and results accordingly.

8.     Authors did not evaluate if extracts are exactly inside the emulsion not in the continuous phase or vice versa. How are you sure that loading happened inside globules?

Did you do any UV specs?

9.     Mention globule sizes for all emulsions.

10.  “Both the ingredients with high antioxidant potential and the physical properties of  the emulsion increase the success rate of the product, as a pleasant component of the daily routine that, easily entered into the habit of frequent care, also ensure the active ingredients’ real chances of protection.”

 Which 2 you are referring to?

Authors did not perform any antioxidant studies so restrict your conclusion to the formulation part.

11.  “RV and FA were purchased as pure substances (>90%) obtained from plant resources”

Which country and what plant species.

12.  All machines should be as follows, Name, manufacturer and city country name. Software version.

13.  Mention number of repeats for all figures. If its one mention single measurement. 

14. Why did you skip C formulation for antimicrobial studies. Is there any specific reason. 

Reviewer 2 Report

The article describes the development and evaluation of a cosmetic using natural ingredients. The text is well-written and has a good depth on the tests performed.

Some minor comments are:

- Methods: the citation of the equipment used is not consistent. Some equipment have brand, model and local of production, and others don't. Then, please add complete information of all equipment and raw materials used.

- Table 2 lacks some info. A table should be self-explanatory, so please add more information on the stability study conditions (as footnote). Also explain why some info is missing (for example, conductivity), and why some pH results have only one value while others have two.

- Figure 2 has a typo: positive is with an s, not with a z.

- Figures 3-7: caption needs to be improved and more clear.

- Table 3: caption needs to be improved and more clear.

- It is never clear how long the stability studies were carried out. Apparently only at the beginning? So would this be a feasibility study, instead of a study study? Stability studies need to run over a period of time, under accelerated and long-term conditions. The text currently does not explain anything about that. So please add a bit on stability studies and why your work did not run any of those storage conditions.

- Delete item 6 (Patent).

Reviewer 3 Report

Let me make some remarks.

The authors investigate the properties of "... emulsions that contain as active components resveratrol (RV) (3%), ferulic acid (FA) (0.5%) and 1mL vegetal extract of Saffron".  1ml per what? Perhaps, you mean 100 ml? Could you clarify this?

Section 5. Conclusion is not too similar to the title of the paper. Perhaps the meaning of research and further plans for their use should be clarified.

English level needs to be edited.

Reviewer 4 Report

Turcov and co-workers presented an original paper for Pharmaceutics. Authors analysed emulsion based on resveratrol, ferulic acid and alcoholic extract Saffron considering some fundamental criteria for a new marketed product. Although the aim is interesting, the manuscript needs substantial revisions and it is not up to par to this journal. In detail:

- methods are poorly described; microscopies images have low resolution;

- emulsion sizes analyses are completely missing;

- table 2 is repetitive, unclear in the section referred to pH and lacking important information;

- it is not clear why figure 2 reports only two type of samples and why the text is referring on a "number of measured colony-forming units per gram of sample" that is not reported in figure;

- results and discussion must be improved;

- conclusions are too exagerates, such an example authors speaks about "high degree of stability to the action of physical stimuli" but some formulation show a separation phase. 

For this and other reasons it is opinion of this reviewer that it should be REJECTED in the present form. 

Round 2

Reviewer 1 Report

The authors addressed my concerns. Disclose pending patents/IP rights if any in the conflicts statement. 

Reviewer 4 Report

the authors have improved the quality of the manuscript. It can be accepted in the current form.